# Inhibition of LncRNA Kcnq1ot1 suppresses hypoxia-induced pyroptosis of H9C2 cells by regulating miR-27b-3p

Yingjie Yang◎, Yanchun Ou◎, Guanlian Mo, Jing Wen, Limin Liang, Shirong Wang, Jinyi Li◎*

Guilin Medical University Affiliated Hospital, Guilin, Guangxi, China

◎ These authors made equal contributions.
* yzrcg@126.com

## Abstract

### Background

Heart failure (HF) is a major cardiovascular disease with high mortality worldwide, whose pathophysiology is multifaceted. Hypoxia has emerged as a critical factor contributing to the progression of heart failure. We aimed to examine the expression and functions of LncRNA Kcnq1ot1 in hypoxia-induced cardiomyocytes in the process of HF.

### Methods

H9C2 cell model was simulated by hypoxia treatment. TUNEL, ELISA, Western Blot and qRT-PCR assay were carried out to evaluate cell pyroptosis, inflammation and dysfunction. Subsequently, we identified the direct downstream target of Kcnq1ot1 by bioinformatics analysis, RNA pull-down, double Luciferase reporter gene and other functional experiments.

### Results

Firstly, Kcnq1ot1 levels was revealed to be upregulated in hypoxia cells than in control cells, and miR-27b-3p showed the opposite trend. And as expected, inhibition of Kcnq1ot1 and overexpression of miR-27b-3p both protected H9C2 against hypoxia-induced pyroptosis, inflammation and dysfunction. Moreover, miR-27b-3p was proved to bind with Kcnq1ot1 and participated in Kcnq1ot1-mediated H9C2 injury under hypoxia by regulating the Wnt3a/β-Catenin/NLRP3 signaling pathway.

### Conclusions

Collectively, our study demonstrated that inhibition of Kcnq1ot1 protected cardiomyocyte against hypoxia-induced injury possibly via sponging miR-27b-3p, which could be useful as biomarkers and therapeutic targets for HF patients.

**Data availability statement:** All data are in the manuscript and/or supporting information files.

**Funding:** This study was supported by the National Natural Science Foundation of China (Grant No. 82160077), the Self-Funded Scientific Research Project of Guangxi Health Department (Grant No. Z20211177) and the General Program of Natural Science Foundation of Guangxi Province of China (Grant No. 2017GXNSFAA198129).

**Competing interests:** The authors have no conflicts of interest to declare.

## Introduction

Heart failure (HF) is a multifaceted clinical syndrome characterized by both systolic and diastolic dysfunction, leading to an imbalance between the demand for and supply of oxygenated blood [1]. This debilitating condition affects around 26 million individuals globally and results in over 1 million hospitalizations annually in the United States and Europe [2]. Furthermore, HF significantly contributes to high morbidity and mortality rates, as it elevates the risk of stroke [3]. Although recent advancements in medical and instrumental therapies have shown promise in improving cardiac function, HF continues to be a leading cause of death worldwide [4]. Current treatment strategies remain limited in their ability to alleviate symptoms and halt disease progression, highlighting the urgent need for a deeper understanding of HF to facilitate the development of innovative therapeutic approaches.

Noncoding RNAs (ncRNAs) are a category of RNA molecules that do not code for proteins. Remarkably, about 98% of the human genome consists of ncRNAs [5], which play crucial regulatory roles within extensive communication networks [6]. Among these, long ncRNAs (lncRNAs) and microRNAs (miRNAs) have recently garnered significant attention due to their vital functions in regulating cell proliferation, differentiation, apoptosis, and migration [7,8]. For instance, silencing of XIST improved cardiac function and survival rate and reduced apoptosis and pyroptosis in septic rats in vivo [9].

Recent studies have highlighted the important role of KCNQ1 overlapping transcript 1 (Kcnq1ot1) in cardiac diseases. Silencing Kcnq1ot1 has been shown to reduce pyroptosis and fibrosis in diabetic cardiomyopathy [10]. Additionally, Kcnq1ot1 influences chromatin structure and Kcnq1 expression during heart development [11]. Knockdown of Kcnq1ot1 has also been demonstrated to offer protection against cell apoptosis in cases of myocardial ischemia/reperfusion injury following acute myocardial infarction (AMI) [12].

LncRNAs have been shown to influence messenger RNA (mRNA) expression through a competing endogenous RNA (ceRNA) regulatory network at the post-transcriptional level [13]. This ceRNA mechanism involves lncRNAs competing with mRNAs for binding to miRNAs, thereby reducing the inhibitory effects of miRNAs on mRNA targets [14]. The ceRNA network involving Kcnq1ot1 has been extensively studied. For example, kcnq1ot1 promotes macrophage lipid accumulation and accelerates the development of atherosclerosis through the miR-452-3p/HDAC3/ABCA1 pathway [15].

In our prior research, we investigated the alleviation of atrial fibrosis (AF) in atrial fibrillation rats by Wnt3a-targeted regulation of the signaling of Wnt/β-Catenin through miR-27b-3p overexpression [16].

Given these findings, we aim to examine the function and the potential ceRNA role of lncRNAKcnq1ot1 and miR-27b-3p in hypoxia-caused cardiomyocytes.

## Materials and methods

### Cell culture and treatment

H9C2 cells (Pricella, China) were propagated in the dulbecco's modified eagle medium (DMEM) culture medium (Gibco, China) with 10% FBS (Excell Bio, China)

and 1% Penicillin-Streptomycinas (P/S, Beyotime, China) as supplements at 37°C supplied with 5% $CO_2$. To make hypoxic injury, H9C2 cells were incubated in a hypoxic incubator containing 5% $CO_2$, 94% $N_2$ and 1% $O_2$ for 12h then cultured with the normal medium in a normoxic incubator for 2h.

## Cell transfection

si-Kcnq1ot1, miR-27b-3p inhibitor, miR-27b-3p mimic, and their respective negative controls (NC) were purchased from Ribobio (Guangzhou, China). An equal amount of H9C2 cells were seeded in a 6-well plate and Lipofectamine 3000 transfection reagent (ThermoFisher, USA) and used for cell transfection. The transfection time was 24h.

## RNA extraction and quantitative real-time PCR (qRT-PCR)

Based on the supplier's instruction, total RNA samples were extracted with Cell/Tissue Total RNA Isolation Kit V2 (RC112, Vazyme, China) and used for generating cDNA with HiScript III 1st Strand cDNA Synthesis Kit (R312, Vazyme, China). Taq Pro Universal SYBR qPCR Master Mix (Q712, Vazyme, China) was acquired for qRT-PCR. Relative gene expression was standardized to GAPDH or U6 after calculating with $2^{-\Delta\Delta CT}$ method.

The primer sequence is as follows: Kcnq1ot1: forward: 5'-TATGGCAAAACCCGGATGGG-3'; reverse: 5'-TGGCTAGTCCCGATAGGGTG-3'. miR-27b-3p: forward: 5'- GCGCGTTCACAGTGGCTAAG-3'; reverse: 5'-AGTGCAGGGTCCGAGGTATT-3'. GAPDH: forward: 5'-CCCTTAAGAGGGATGCTGCC-3'; reverse: 5'-TACGGCCAAATCCGTTCACA-3'. U6: forward: 5'-GGAGACACGCAAACGGAAG-3'; reverse: 5'-AGTGCAGGGTCCGAGGTATT-3'.

## Western blot (WB)

Total protein in cells was extracted by using RIPA lysis buffer (Beyotime, China). The proteins were separated by SDS PAGE and were transferred onto PVDF membranes. After blocking for 1h at room temperature with 5% non-fat milk in Tris buffered saline-0.01% Tween 20 (Biosharp, China), the membranes were incubated with primary antibodies at 4°C overnight for detection of Wnt3a (26744–1-AP, Proteintech, China), β-Catenin (A19657, ABclonal, China), p-β-Catenin (AP1076, ABclonal, China), NLRP3 (A5652, ABclonal, China), Caspase-1 (31020–1-AP, Proteintech, China), Fibronectin (66042–1-Ig, Proteintech, China), Collagen III (ab184993, Abcam, UK) and β-tubulin(66240–1-Ig, Proteintech, China). After three washes with Tris buffered saline-0.01% Tween 20, the membranes were incubated with the corresponding secondary antibody for 1h at room temperature. The signals were developed by using the chemiluminescence detection kit (NCM Biotech, China), and the blots were finally monitored via ECL detection system (JP-K600, China).

## Enzyme-linked immunosorbent assay (ELISA)

Concentrations of interleukin (IL)-1β and IL-18 in the culture medium were determined using ELISA kits (COIBOBIO, China). The optical value of each well was determined at 450nm using a microplate reader and converted to the corresponding concentration according to the standard curve.

## TdT-mediated dUTP Nick-End Labeling (TUNEL) assay

TUNEL Apoptosis Assay Kit was produced by Beyotime (C1088, China) and acquired for assessing the cell samples in accordance with the user manual. After washing in phosphate buffer saline (PBS), cell samples were probed with TUNEL detection kit after permeabilization. Cell nuclei were counterstained with 4',6-diamidino-2-phenylindole (DAPI), then estimated with optical microscopy.

## Immunofluorescence (IF) staining

H9C2 cells or primary cardiomyocytes were fixed, permeabilized and blocked. Then, the cells were incubated with GSDMD-N antibody (CST, USA) at 4°C overnight and incubated with Cy3-conjugated secondary antibody (Yeasen, China)

in a dark room at 37°Cfor 1h. DAPI was used for nucleus staining. Cells were then observed under optical microscopy and representative images were captured.

## Dual-luciferase reporter assays

The wild-type (WT) and mutated (MUT) Kcnq1ot1 fragments covering the miR-27b-3p binding sites were synthesized and inserted into pmirGLO luciferase Vector, termed Kcnq1ot1-Wt/Mut reporter vectors. The H9C2 cells were co-transfected with Kcnq1ot1 WT/MUT and miR-27b-3p mimic/NC. After incubation, cells were collected to measure the firefly and renilla luciferase activity using the Luciferase Assay kit (Beyotime, China). Relative luciferase activity was normalized using the renilla luciferase activity.

## RNA pull down

The RNA extracts from H9C2 samples were cultivated with the biotin-labelled Kcnq1ot1 probe (Kcnq1ot1 biotin probe) or Kcnq1ot1 no-biotin probe as NC, along with the magnetic beads for 1h. The pulled-down complex was monitored by qRT-PCR method.

## RNA immunoprecipitation (RIP) assay

The H9C2 samples in RIP lysis buffer were harvested and mixed with the magnetic beads-bound specific antibody to Argonaute2 (Ago2) or Immunoglobulin G (IgG) as NC in the RIP buffer for 1h. Precipitated RNAs were subjected to qRT-PCR analysis for confirming the presence of binding sites.

## Statistics

Data were analyzed and plotted using Graphpad 9 (Version 9.4). AI 2023 collates the graph. All data were represented by means±SD, and the statistical difference between groups was tested by t-test or one-way test, and the p value<0.05 was considered as significant difference.

## Results

### Hypoxia induces inflammation, myocardial remodeling, and pyroptosis of cells

The Wnt signaling pathway plays a crucial role in cardiac development, cell proliferation, and differentiation [17,18]. In HF, abnormal activation of the Wnt signaling pathway may lead to pathological remodeling of myocardial cells, including fibrosis [19], apoptosis [20] and pyroptosis [21]. As expected, compared to the normoxia control group, we observed a significant activation of the Wnt signaling pathway in H9C2 cells in the hypoxia group, which was evidenced by an increase in Wnt3a, a decrease in p-β-Catenin, and an increase in β-Catenin at the protein levels (Fig 1A). Studies have demonstrated that the activation of the Wnt signaling pathway may influence the activation of the NLRP3 inflammasome, which subsequently activates Caspase-1, leading to the production of IL-1β and IL-18, thereby triggering an inflammatory response [22,23]. In our study, we measured the increased levels of NLRP3, Caspase-1, IL-1β, and IL-18 in hypoxic cells using qRT-PCR, WB, or ELISA (Fig 1B-C), proving the activation of this pathway in hypoxia induced cardiomyocytes. Fibronectin and Collagen III, as critical components of the extracellular matrix (ECM), also play significant roles in the onset and progression of HF [24]. They influence the structure and function of the heart by participating in processes such as myocardial remodeling, inflammatory responses, and fibrosis [25]. In our study, we also assessed the increased levels of them in hypoxic cells using qRT-PCR and WB (Fig 1D). Myocardial pyroptosis is an important contributor to the onset and progression of heart failure, as it promotes inflammatory responses, leads to the release of cytokines, and facilitates myocardial remodeling [26,27]. Using TUNEL staining and IF for GSDMD-N, we measured the levels of pyroptosis and found that they were elevated in hypoxic H9C2 cells (Fig 1E–F). These data collectively indicated that hypoxia triggered

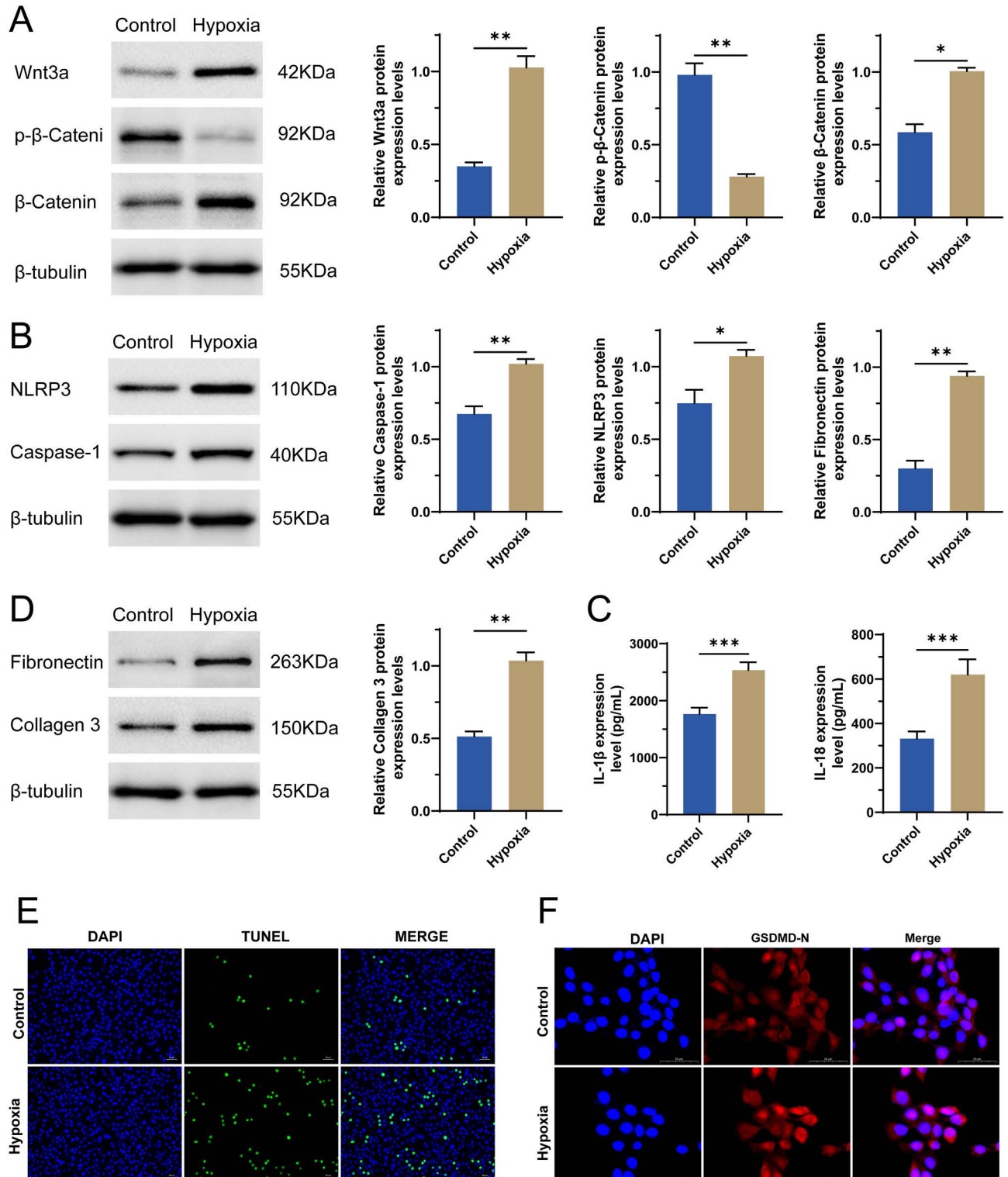

**Fig 1. Effect of hypoxia on H9C2 cells.** (a) Protein levels of Wnt3a, p-β-Catenin and β-Catenin were detected by WB. (b) Protein levels of NLRP3 and Caspase-1 were detected by WB. (c) Levels of IL-1β and IL-18 were detected by ELISA. (d) Protein levels of Fibronectin and Collagen III were detected by WB. (e) TUNEL staining. (f) IF for GSDMD-N in the normoxia control group and hypoxia group. n = 3, *$p < 0.05$, **$p < 0.01$, ***$p < 0.001$.

inflammatory responses, myocardial remodeling, and pyroptosis in H9C2 cells, potentially through the Wnt3a/β-Catenin/NLRP3 signaling pathway.

## Inhibition of lncRNA Kcnq1ot1 attenuates hypoxia-induced cell damage

To start with, we observed that the level of Kcnq1ot1 was significantly increased in hypoxia group as compared to the control group (Fig 2A). To further explore the functional effects of Kcnq1ot1 on hypoxia-injured H9C2 cells, si-Kcnq1ot1 or si-NC was transfected into H9C2 cells, and the si-Kcnq1ot1 was successfully silenced (Fig 2B). Further functional assay results showed that cell damages triggered by hypoxia were attenuated when Kcnq1ot1 was silenced. As compared to the Hypoxia+si-NC group, the Wnt signaling pathway (Fig 2C–D), inflammatory responses (Fig 2E–G), ECM components (Fig 2H–I), and cell pyroptosis (Fig 2J–K) were decreased in the Hypoxia+si-Kcnq1ot1 group. These data collectively indicated that inhibition of Kcnq1ot1 could attenuates hypoxia-induced cell damage.

## miR-27b-3p attenuates hypoxia-induced cell damage

In our prior research, we investigated the alleviation of AF in atrial fibrillation rats by Wnt3a-targeted regulation of the signaling of Wnt/β-Catenin through miR-27b-3p overexpression [16]. Therefore, we also want to verify the role of miR-27b-3p in this hypoxic cell model. We firstly observed that the level of miR-27b-3p was significantly decreased in hypoxia group as compared to the control group (Fig 3A), and miR-27b-3p mimic or NC was successfully transfected into H9C2 cells (Fig 3B). Further functional assay results showed that cell damages introduced by hypoxia were attenuated when Kcnq1ot1 was silenced. As compared to the NC group, the Wnt signaling pathway (Fig 3C–D), inflammatory responses (Fig 3E–G), ECM components (Fig 3H–I), and cell pyroptosis (Fig 3J–K) were decreased in the miR-27b-3p mimic group. These data collectively indicated that miR-27b-3p could attenuates hypoxia-induced cell damage, similar to Kcnq1ot1 silencing.

## LncRNA Kcnq1ot1 works as a sponge for miR-27b-3p

To analyse the molecular mechanism, the subcellular localization of Kcnq1ot1 was predicted through the LncATLAS website (http://lncatlas.crg.eu/) [28], predicting that lncRNA MIAT might be located in the nucleus (Fig 4A). And a nuclear/cytosol fractionation assay confirmed that Kcnq1ot1 was mainly expressed in the nucleus (Fig 4B). In order to reveal whether Kcnq1ot1 functioned to H9C2 cells in miR-27b-3p mediated signaling, the regulatory relationship between them was studied. qRT-PCR data showed that transfection of cells with si-Kcnq1ot1 significantly up-regulated miR-27b-3p expression, when compared to si-NC group (Fig 4C), which indicated that miR-27b-3p was negatively regulated by Kcnq1ot1. Then, bioinformatics analysis showed that Kcnq1ot1 contains a binding site of miR-27b-3p (Fig 4D).

To validate whether Kcnq1ot1 could directly bind with miR-27b-3p, the affinity between them was researched by RNA pull down assay in H9C2 cells, and the results demonstrated that miR-27b-3p was enriched in Kcnq1ot1 biotin group (Fig 4E), suggesting that miR-27b-3p might bind with Kcnq1ot1. For the further exploration of the regulation of Kcnq1ot1 on miR-27b-3p, luciferase reporter assay was performed. The luciferase activity of Kcnq1ot1-WT reporter was suppressed by enforced expression of miR-27b-3p, while that of Kcnq1ot1-MUT reporter was not affected (Fig 4F). Our previous research has validated the interaction between miR-27b-3p and Wnt3a using a firefly luciferase reporter gene [16]. Therefore, based on the above data, we speculate that Kcnq1ot1 competes with miR-27b-3p to enhance the expression of Wnt3a.

## Inhibition of Kcnq1ot1 protects H9C2 cells against hypoxia-induced cell damage via up-regulation of miR-27b-3p

In order to validate the abovementioned hypothesis, an inhibitor specific for miR-27b-3p was transfected into H9C2 cells. The expression of miR-27b-3p was significantly decreased in miR-27b-3p inhibitor group than that in the NC group (Fig 5A). Of note, the Wnt signaling pathway (Fig 5B–C), inflammatory responses (Fig 5D–F), ECM components (Fig 5G–H), and cell pyroptosis (Fig 5I–J) were increased in the Hypoxia+si-Kcnq1ot1+miR-27b-3p inhibitor group, than

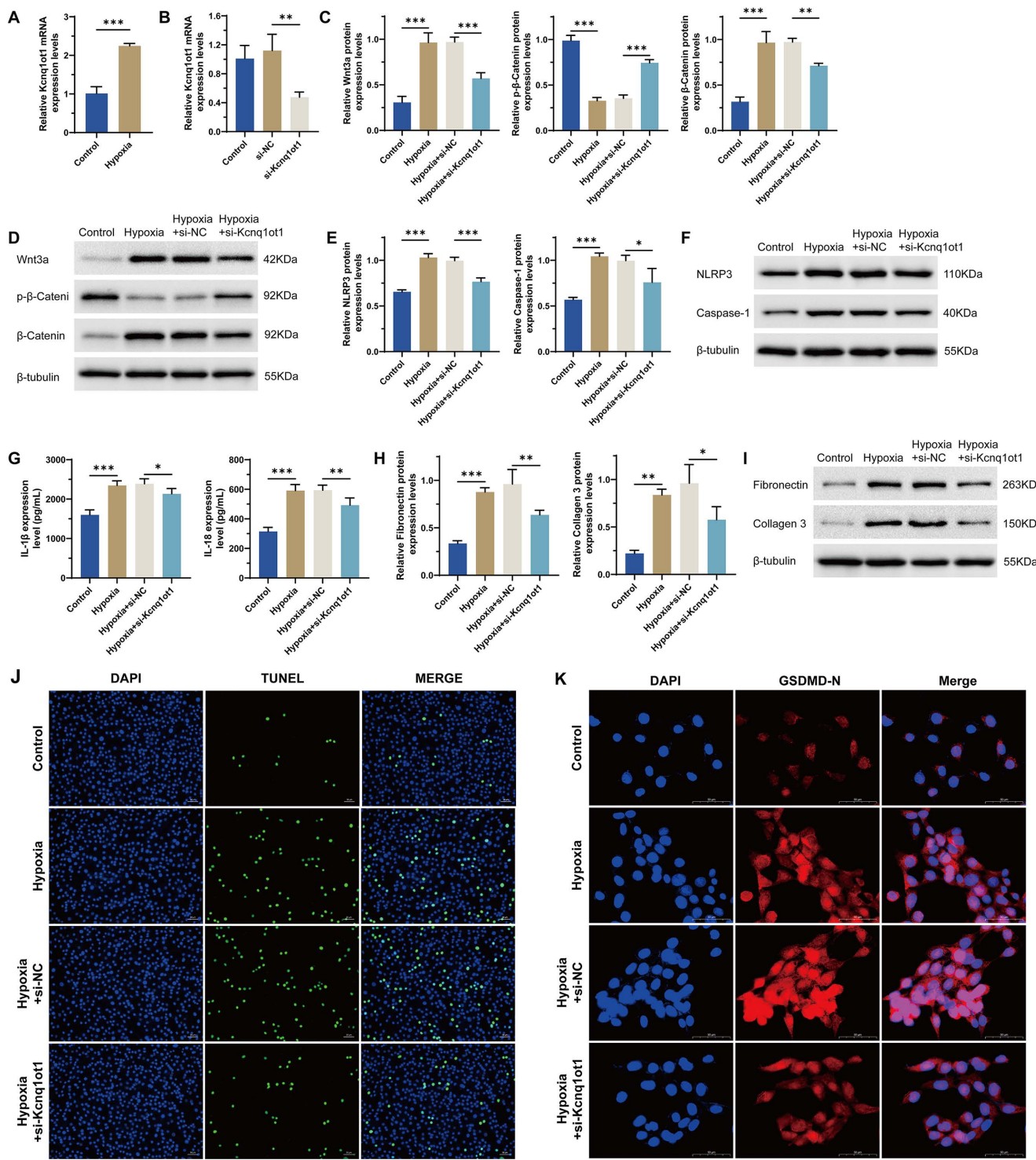

**Fig 2. Effect of Kcnq1ot1 inhibiting on hypoxia-injured H9C2 cells.** (a) RNA level of Kcnq1ot1 by qRT-PCR in the normoxia control group and hypoxia group. (b) RNA levels of Kcnq1ot1 by qRT-PCR in the si-NC group and si-Kcnq1ot1 group. (c) mRNA level of Wnt3a, p-β-Catenin and β-Catenin by qRT-PCR. (d) Protein levels of NLRP3 and Caspase-1 by WB. (e) mRNA level of NLRP3 and Caspase-1 by qRT-PCR. (f) Protein level of NLRP3 and Caspase-1 by WB. (g) Levels of IL-1β and IL-18 by ELISA. (h) mRNA levels of Fibronectin and Collagen III by qRT-PCR. (i) Protein levels of Fibronectin and Collagen III by WB. (j) TUNEL staining. (k) IF for GSDMD-N in the Hypoxia+si-NC group and the Hypoxia+si-Kcnq1ot1 group. n = 3, *p < 0.05, **p < 0.01, ***p < 0.001.

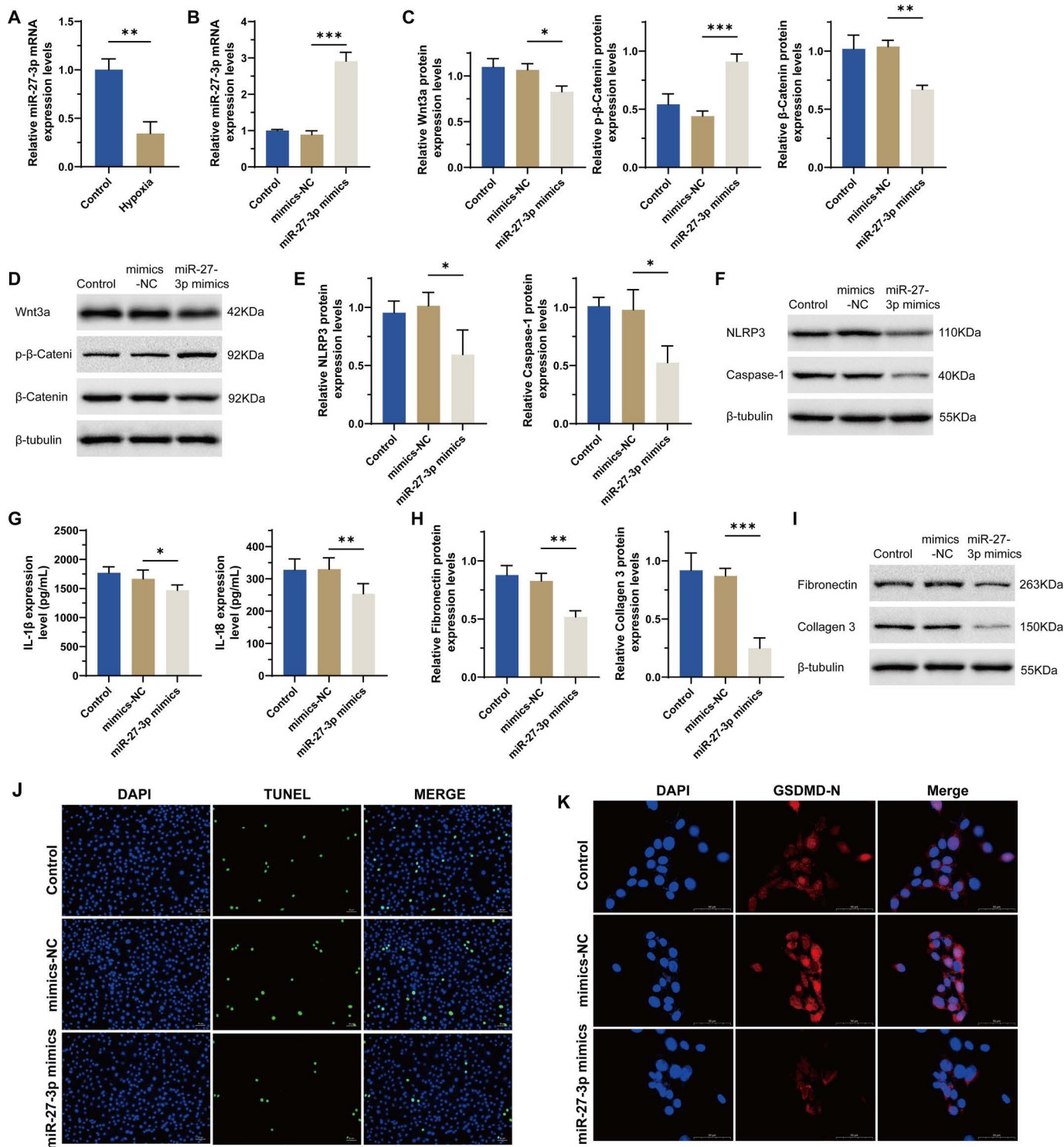

**Fig 3. Effect of miR-27b-3p on hypoxia-injured H9C2 cells.** (a) RNA levels of miR-27b-3p by qRT-PCR in the normoxia control group and hypoxia group. (b) RNA levels of miR-27b-3p by qRT-PCR. (c) mRNA levels of Wnt3a, p-β-Catenin and β-Catenin by qRT-PCR. (d) Protein levels of NLRP3 and Caspase-1 by WB. (e) mRNA levels of NLRP3 and Caspase-1 by qRT-PCR. (f) Protein levels of NLRP3 and Caspase-1 by WB. (g) Levels of IL-1β and IL-18 by ELISA. (h) mRNA levels of Fibronectin and Collagen III by qRT-PCR. (i) Protein levels of Fibronectin and Collagen III by WB. (j) TUNEL staining. (k) IF for GSDMD-N in the mimic-NC group and the miR-27b-3p mimic group. n = 3, *$p < 0.05$, **$p < 0.01$, ***$p < 0.001$.

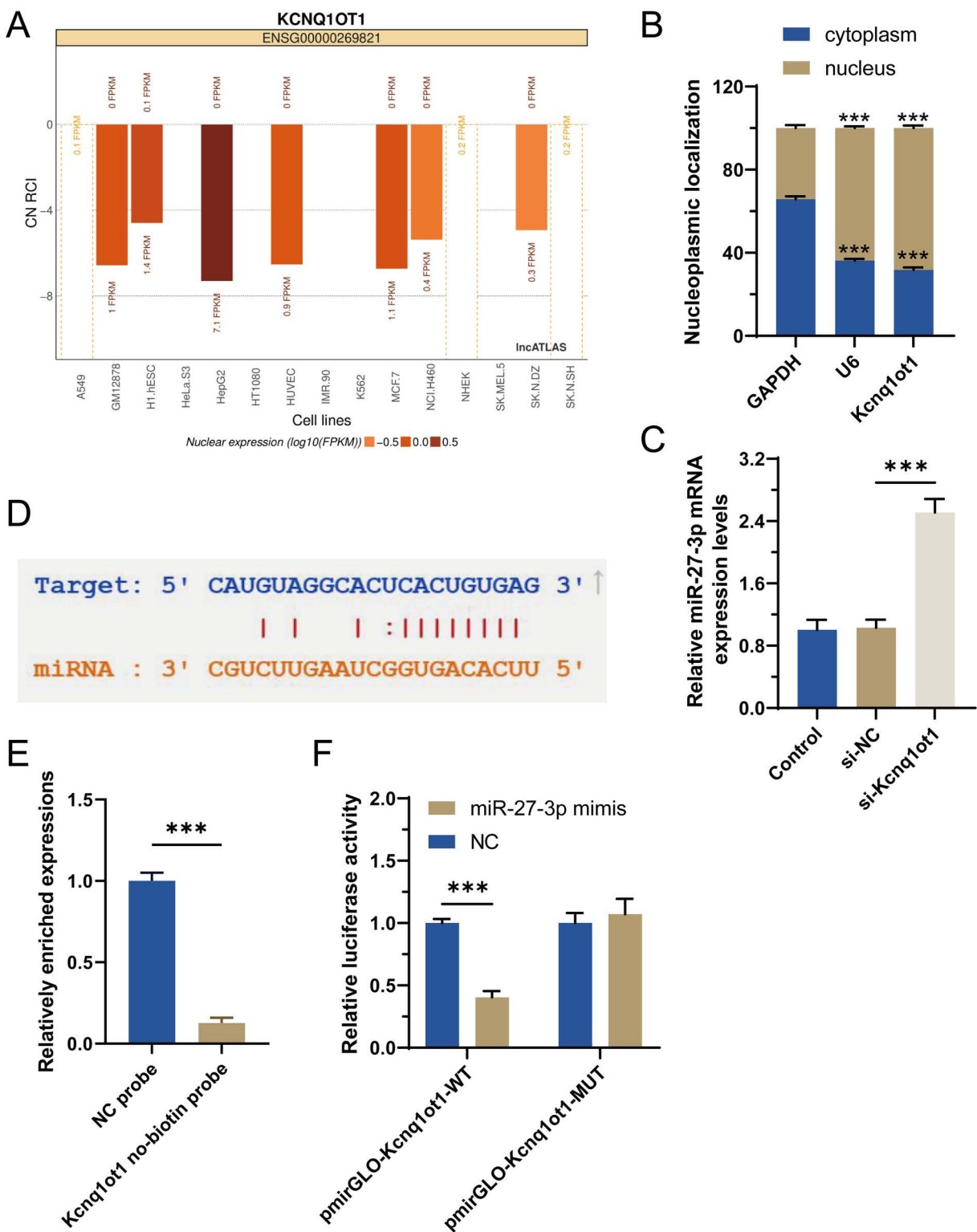

**Fig 4. Effect of Kcnq1ot1 on miR-27b-3p. (a) Subcellular localization of Kcnq1ot1 predicted through the LncATLAS website.** (b) The possible major expression of Kcnq1ot1 in the nucleus of cardiomyocytes via nuclear/cytosolic fractionation assay. (c) RNA levels of miR-27b-3p by qRT-PCR in the si-NC group and si-Kcnq1ot1 group. (d) The binding sites between Kcnq1ot1 and miR-27b-3p utilizing starBase. (e) RNA pull down assay of the binding between Kcnq1ot1 and miR-27b-3p. (f) Luciferase reporter assay of the combination between Kcnq1ot1 and miR-27b-3p. n=3, *p<0.05, **p<0.01, ***p<0.001.

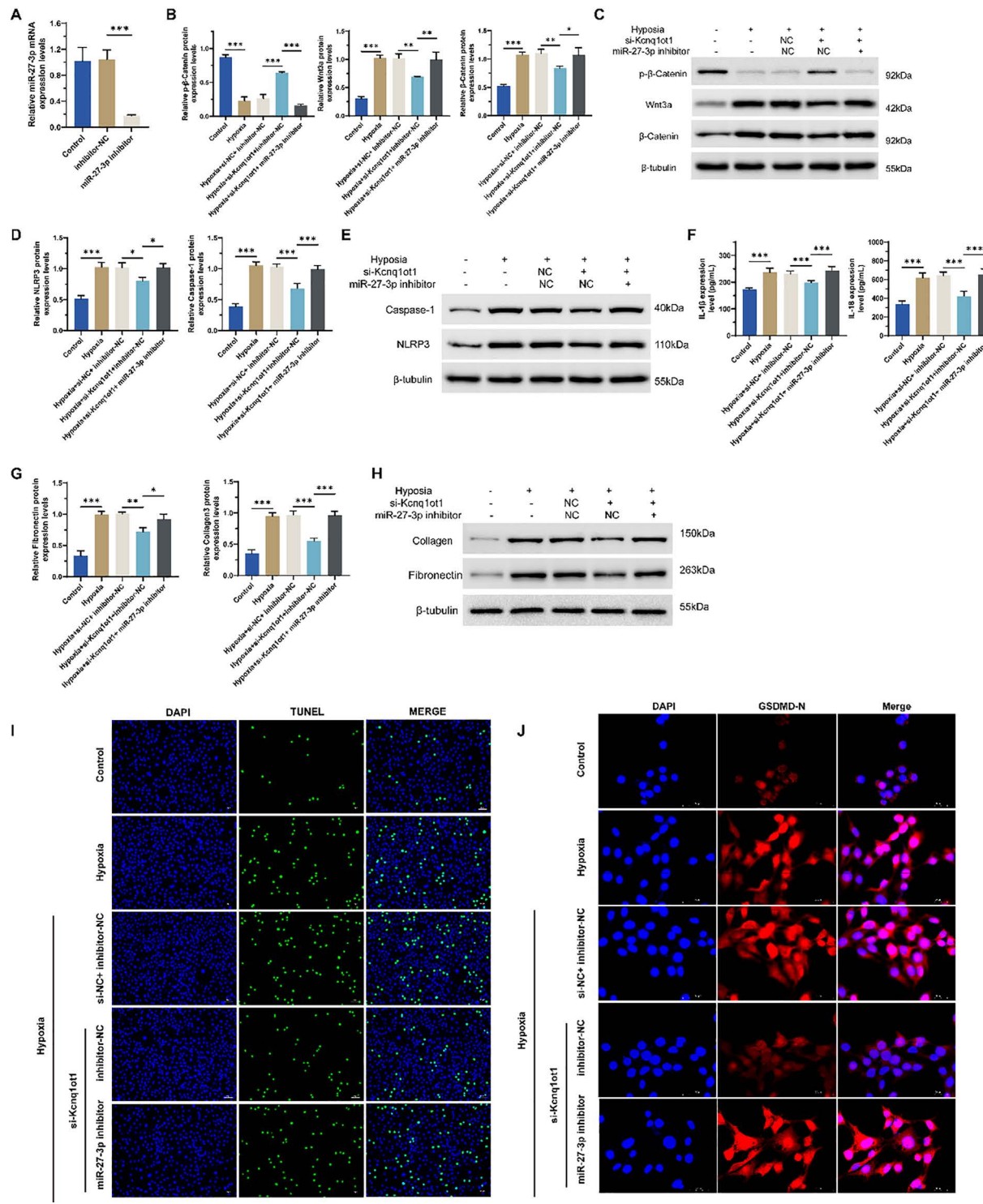

**Fig 5. Effect of miR-27b-3p suppression together with Kcnq1ot1 inhibiting on hypoxia-injured H9C2 cells.** (a) RNA levels of miR-27b-3p by qRT-PCR in the inhibitor-NC group and miR-27b-3p inhibitor group. (b) mRNA levels of Wnt3a, p-β-Catenin and β-Catenin by qRT-PCR. (c) Protein levels of NLRP3 and Caspase-1 by WB. (d) mRNA levels of NLRP3 and Caspase-1 by qRT-PCR. (e) Protein levels of NLRP3 and Caspase-1 by WB. (f) Levels of IL-1β and IL-18 by ELISA. (g) mRNA levels of Fibronectin and Collagen III by qRT-PCR. (h) Protein levels of Fibronectin and Collagen III by WB. (i) TUNEL staining. (j) IF for GSDMD-N in the Hypoxia+si-Kcnq1ot1+inhibitor-NC group and the Hypoxia+si-Kcnq1ot1+miR-27b-3p inhibitor group. n = 3, *$p < 0.05$, **$p < 0.01$, ***$p < 0.001$.

those in Hypoxia+si-Kcnq1ot1+inhibitor-NC group. Collectively, it seems that the effects of Kcnq1ot1 are impeded when miR-27b-3p is knocked down. Thus, we preliminarily conclude that inhibition of Kcnq1ot1 promotes the activation of Wnt3a/β-Catenin/NLRP3 signaling pathways and cell pyroptosis possibly via regulating miR-27b-3p.

## Discussion

Hypoxia could result in myocardial cell injuries, which further lead to the initiation of several cardiovascular diseases, including HF and AMI. It is significant to find key molecules that protect cardiomyocytes from hypoxia-caused injury. The hypothesis of the present study was that inhibition of Kcnq1ot1 could protect H9C2 cells against hypoxia-introduced cell injury. Therefore, based on previous research, predictions from the database, interaction experiment and functional assay, this study found that Kcnq1ot1 was upregulated in hypoxia cells and miR-27b-3p showed the opposite trend, inhibition of Kcnq1ot1 and overexpression of miR-27b-3p both protected H9C2 against hypoxia-induced pyroptosis, inflammation and dysfunction. Moreover, miR-27b-3p was confirmed as the target of Kcnq1ot1 and participated in Kcnq1ot1-mediated H9C2 injury under hypoxia by regulating the Wnt3a/β-Catenin/NLRP3 signaling pathway (**Fig 6**).

A growing number of lncRNAs have been linked to various kinds of cardiovascular diseases. For instance, increased expression of LncRNA Kcna2 Antisense RNA led to an increased incidence of ventricular arrhythmias in association with heart failure [29]. LncRNA UCA1 was able to promote the progression of cardiac hypertrophy, a condition associated with a series of cardiovascular diseases, including heart failure [30]. In our study, we explored the functional role of Kcnq1ot1, in hypoxia-injured H9C2 cells, aiming to evaluate the importance of Kcnq1ot1 in HF caused by myocardial infarction. It is worth noting that in non-cardiovascular fields, we have also found examples of Kcnq1ot1 activating the Wnt/β-catenin signaling pathway [31], as well as examples of Kcnq1ot1 promoting cell pyroptosis by inhibiting miRNA and upregulating

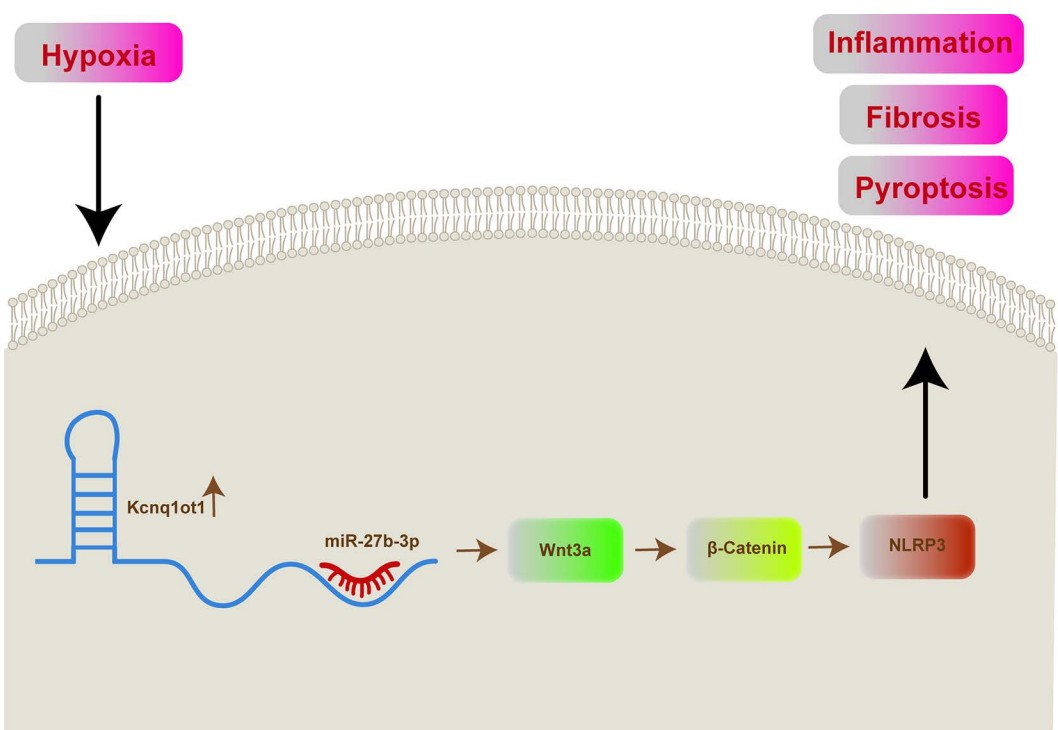

**Fig 6. Schematic of Kcng1ot1/miR-27b-3p/Wnt3a axis under hypoxia.** Kcnq1ot1 sequesters miR-27b-3p to activate Wnt3a/β-Catenin/NLRP3-mediated pyroptosis, fibrosis and inflammation.

NLRP3 [32], which had something in common with our study. In cardiovascular fields, Li et al have revealed the role of Kcnq1ot1 to induce H9C2 apoptosis in myocardial ischemia/reperfusion (I/R), which is a major cause for AMI [33], while our study focused on molecular mechanism of hypoxia-induced cell damage during the process of HF. The following limitations should be taken into consideration as interpreting our findings. We performed our study in H9C2, but it may not tell the whole story concerning the exact mechanisms of Kcnq1ot1, which could not be extrapolated to primary cell level and animal level, even if they share some similarities. Moreover, our findings may be useful as biomarkers and therapeutic targets for HF, but further studies in rats and clinical trials are warranted to dissect its mechanisms and clinical application.

## Conclusion

Collectively, this paper discussed the ceRNA role of Kcnq1ot1 in hypoxia-induced cardiomyocytes. Kcnq1ot1 elevated Wnt3a expression to facilitate cardiomyocyte injury via sequestering miR-27b-3p, shedding a new light on the pathogenesis of HF.

## Supporting information

**S1 File. Raw images.**
(PDF)

## Author contributions

**Conceptualization:** Yingjie Yang, Jinyi Li.

**Data curation:** Guanlian Mo, Jing Wen, Limin Liang, Shirong Wang.

**Formal analysis:** Yingjie Yang, Yanchun Ou, Guanlian Mo, Jing Wen, Limin Liang, Shirong Wang.

**Investigation:** Guanlian Mo, Jing Wen, Limin Liang, Shirong Wang, Jinyi Li.

**Writing – original draft:** Yingjie Yang, Yanchun Ou, Guanlian Mo, Jing Wen, Limin Liang, Shirong Wang.

**Writing – review & editing:** Yingjie Yang, Yanchun Ou, Jinyi Li.

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
