## [Decision Letter · Decision Letter 0]

21 Jul 2025

PONE-D-25-11818Inhibition of LncRNA Kcnq1ot1 suppresses hypoxia-induced pyroptosis of H9C2 cells by regulating miR-27b-3pPLOS ONE

Dear Dr. Li,

Thank you for submitting your manuscript to PLOS ONE. After careful consideration, we feel that it has merit but does not fully meet PLOS ONE’s publication criteria as it currently stands. Therefore, we invite you to submit a revised version of the manuscript that addresses the points raised during the review process.

**ACADEMIC EDITOR: ** The Reviewer 1 suggested that external experimant should be made. Please note that.

We look forward to receiving your revised manuscript.

Kind regards,

Zhanzhan Li

Academic Editor

PLOS ONE

Journal Requirements:

 "This study was supported by the National Natural Science Foundation of China (Grant No. 82160077), the Self-Funded Scientific Research Project of Guangxi Health Department (Grant No. Z20211177) and the General Program of Natural Science Foundation of Guangxi Province of China (Grant No. 2017GXNSFAA198129)."      

4. PLOS ONE now requires that authors provide the original uncropped and unadjusted images underlying all blot or gel results reported in a submission’s figures or Supporting Information files. This policy and the journal’s other requirements for blot/gel reporting and figure preparation are described in detail at https://journals.plos.org/plosone/s/figures#loc-blot-and-gel-reporting-requirements and https://journals.plos.org/plosone/s/figures#loc-preparing-figures-from-image-files. When you submit your revised manuscript, please ensure that your figures adhere fully to these guidelines and provide the original underlying images for all blot or gel data reported in your submission. See the following link for instructions on providing the original image data: https://journals.plos.org/plosone/s/figures#loc-original-images-for-blots-and-gels .    

Reviewers' comments:

Reviewer's Responses to Questions

**Comments to the Author**

1. Is the manuscript technically sound, and do the data support the conclusions?

Reviewer #1: Partly

Reviewer #2: Yes

2. Has the statistical analysis been performed appropriately and rigorously? 

Reviewer #1: Yes

Reviewer #2: I Don't Know

3. Have the authors made all data underlying the findings in their manuscript fully available?

Reviewer #1: Yes

Reviewer #2: Yes

4. Is the manuscript presented in an intelligible fashion and written in standard English?

Reviewer #1: Yes

Reviewer #2: Yes

5. Review Comments to the Author

Reviewer #1: In the provided manuscript, authors showed that Kcnq1ot1 contains a binding site of miR-27b-3p using bioinformatic analysis. They also performed RNA pull downs assays to showcase the interaction between the LncRNA and miR-27b-3p. However, it would be strongly suggested to perform additional experiments like EMSAs to demonstrate the direct interaction between the two RNA molecules. It would also be suggested that both WT and mutated sequences be tested to demonstrate specific interactions between the two RNA molecules in question. These experiments deemed critical to support the conclusion in the study that inhibition of Kcnq1ot1 protected cardiomyocyte against hypoxia-induced injury possibly via sponging miR-27b-3p.

Reviewer #2: In line 110-113 : the authors mentioned two reverse primers for GAPD and U6 with no information about forward primer, How?!

Authors should define clearly the laboratory where this research done.

Authors should introduce results in the form of graphs and/or tables to be more obvious.

6. PLOS authors have the option to publish the peer review history of their article (what does this mean? ). If published, this will include your full peer review and any attached files.

**Do you want your identity to be public for this peer review?** For information about this choice, including consent withdrawal, please see our Privacy Policy .

Reviewer #1: No

Reviewer #2: **Yes: ** Sarah Samy Abdelghany

---

## [Author Response · Author response to Decision Letter 1]

18 Aug 2025

Dear editor:

Thank you for your Email on 21-July-2025. We truly appreciate the constructive comments and suggestions from you and reviewers to our manuscript (Submission ID: PONE-D-25-11818). Based on the comments, we revised our manuscript and hope the modifications meet your expectations. The changes are in the revised manuscript. The details of our responses are as follows.

Editor Comments:

ACADEMIC EDITOR: The Reviewer 1 suggested that external experimant should be made. Please note that.

Response: Thank you for your reminder. We have reviewed the formatting of the manuscript, and revised the manuscript as requested. The EMSA assays are valuable approaches. EMSA is mainly used to detect interactions between proteins and RNA. However the interaction between Kcnq1ot1 and miR-27b-3p is generally validated by Luciferase assays and RNA pull-down assays. Our data already demonstrate the specificity of this interaction.

"This study was supported by the National Natural Science Foundation of China (Grant No. 82160077), the Self-Funded Scientific Research Project of Guangxi Health Department (Grant No. Z20211177) and the General Program of Natural Science Foundation of Guangxi Province of China (Grant No. 2017GXNSFAA198129)."

Response: Thank you for your comments. The funders had no role in study design, data collection and analysis, decision to publish, or preparation of the manuscript.

Response: Thank you for your comments. All data are in the manuscript and/or supporting information files.

Response: Thank you for your reminder. We have organized the original data for all the gel images and submitted it as "S1_raw_images". While, we have indicated in the cover letter that the uncropped original images are provided in the supporting information.

Response: Thank you for your reminder.

Reviewers' comments:

Reviewer #1:

In the provided manuscript, authors showed that Kcnq1ot1 contains a binding site of miR-27b-3p using bioinformatic analysis. They also performed RNA pull downs assays to showcase the interaction between the LncRNA and miR-27b-3p. However, it would be strongly suggested to perform additional experiments like EMSAs to demonstrate the direct interaction between the two RNA molecules. It would also be suggested that both WT and mutated sequences be tested to demonstrate specific interactions between the two RNA molecules in question. These experiments deemed critical to support the conclusion in the study that inhibition of Kcnq1ot1 protected cardiomyocyte against hypoxia-induced injury possibly via sponging miR-27b-3p.

Response: We thank you for highlighting the importance of rigorously validating the Kcnq1ot1-miR-27b-3p interaction. While EMSA assays are valuable approaches. EMSA is mainly used to detect interactions between proteins and RNA. However the interaction between Kcnq1ot1 and miR-27b-3p is generally validated by Luciferase assays and RNA pull-down assays. Our data already demonstrate the specificity of this interaction:

1.Luciferase assays (Figure 4F) showed that miR-27b-3p suppressed the activity of wild-type (WT) Kcnq1ot1 but not the mutant (MUT) construct, confirming sequence-dependent binding.

2.RNA pull-down (Figure 4E) revealed significant enrichment of miR-27b-3p by Kcnq1ot1 probes versus negative controls.

3.Functional rescue experiments (Figure 5) linked Kcnq1ot1 silencing to miR-27b-3p-mediated Wnt3a downregulation and cytoprotection.

We welcome suggestions to further improve the manuscript within our current technical scope. Thank you very much.

Fig.4. Effect of Kcnq1ot1 on miR-27b-3p. (a) Subcellular localization of Kcnq1ot1 predicted through the LncATLAS website. (b) The possible major expression of Kcnq1ot1 in the nucleus of cardiomyocytes via nuclear/cytosolic fractionation assay. (c) RNA level of miR-27b-3p by qRT-PCR in the si-NC group and si-Kcnq1ot1 group. (d) The binding sites between Kcnq1ot1 and miR-27b-3p utilizing starBase. (e) RNA pull down assay of the binding between Kcnq1ot1 and miR-27b-3p. (f) Luciferase reporter assay of the combination between Kcnq1ot1 and miR-27b-3p. n=3, ***p<0.001.

Fig.5. Effect of miR-27b-3p suppression together with Kcnq1ot1 inhibiting on hypoxia-injured H9C2 cells. (a) RNA level of miR-27b-3p by qRT-PCR in the inhibitor-NC group and miR-27b-3p inhibitor group. (b) mRNA level of Wnt3a, p-β-Catenin and β-Catenin by qRT-PCR. (c) Protein level of NLRP3 and Caspase-1 by WB. (d) mRNA level of NLRP3 and Caspase-1 by qRT-PCR. (e) Protein level of NLRP3 and Caspase-1 by WB. (f) Protein level of IL-1β and IL-18 by ELISA. (g) mRNA level of Fibronectin and Collagen III by qRT-PCR. (h) Protein level of Fibronectin and Collagen III by WB. (i) TUNEL staining. (j) IF for GSDMD-N in the Hypoxia+si-Kcnq1ot1+inhibitor-NC group and the Hypoxia+si-Kcnq1ot1+miR-27b-3p inhibitor group. n=3, *p<0.05, **p<0.01, ***p<0.001.

Reviewer #2: 

In line 110-113 : the authors mentioned two reverse primers for GAPD and U6 with no information about forward primer, How?!

Response: We sincerely appreciate your careful attention to the methodological details. The forward primers for GAPDH and U6 were accidentally omitted during manuscript preparation. The complete primer sets are:

GAPDH: forward: 5’-CCCTTAAGAGGGATGCTGCC-3’;

reverse: 5’-TACGGCCAAATCCGTTCACA-3’.

U6: forward: 5’-GGAGACACGCAAACGGAAG-3’;

reverse: 5’-AGTGCAGGGTCCGAGGTATT-3’.

These sequences have been added to the Materials and Methods section (Page 5, Line 112-115).

Authors should define clearly the laboratory where this research done.

Response: Thank you for your feedback. Most of experiments were conducted in research department of our hospital (Guilin Medical University Affiliated Hospital). Due to the limitation of experimental conditions, few experiments were conducted in the Guilin Medical University.

Authors should introduce results in the form of graphs and/or tables to be more obvious.

Response: Thank you for your recognition and valuable comments on our research. We have supplemented the Graphical Abstract Image in the manuscript, and the content is as follows:

Figure 6. Schematic of Kcng1ot1/miR-27b-3p/Wnt3a axis under hypoxia. Kcnq1ot1 sequesters miR-27b-3p to activate Wnt3a/β-Catenin/NLRP3-mediated pyroptosis, fibrosis and inflammation.

---

## [Decision Letter · Decision Letter 1]

8 Sep 2025

Inhibition of LncRNA Kcnq1ot1 suppresses hypoxia-induced pyroptosis of H9C2 cells by regulating miR-27b-3p

PONE-D-25-11818R1

Dear Dr. Li,

We’re pleased to inform you that your manuscript has been judged scientifically suitable for publication and will be formally accepted for publication once it meets all outstanding technical requirements.

Kind regards,

Zhanzhan Li

Academic Editor

PLOS ONE

Additional Editor Comments (optional):

Reviewer #1:

Reviewer #2:

Reviewers' comments:

Reviewer's Responses to Questions

**Comments to the Author**

1. If the authors have adequately addressed your comments raised in a previous round of review and you feel that this manuscript is now acceptable for publication, you may indicate that here to bypass the “Comments to the Author” section, enter your conflict of interest statement in the “Confidential to Editor” section, and submit your "Accept" recommendation.

Reviewer #1: All comments have been addressed

Reviewer #2: All comments have been addressed

2. Is the manuscript technically sound, and do the data support the conclusions?

Reviewer #1: Yes

Reviewer #2: Yes

3. Has the statistical analysis been performed appropriately and rigorously? 

Reviewer #1: I Don't Know

Reviewer #2: Yes

4. Have the authors made all data underlying the findings in their manuscript fully available?

Reviewer #1: Yes

Reviewer #2: Yes

5. Is the manuscript presented in an intelligible fashion and written in standard English?

Reviewer #1: Yes

Reviewer #2: Yes

6. Review Comments to the Author

Reviewer #1: (No Response)

Reviewer #2: (No Response)

7. PLOS authors have the option to publish the peer review history of their article (what does this mean? ). If published, this will include your full peer review and any attached files.

**Do you want your identity to be public for this peer review?** For information about this choice, including consent withdrawal, please see our Privacy Policy .

Reviewer #1: No

Reviewer #2: **Yes: ** Sarah Samy Abd ElGhany

---

## [Editor Report · Acceptance letter]

PONE-D-25-11818R1

PLOS ONE

Dear Dr. Li,

I'm pleased to inform you that your manuscript has been deemed suitable for publication in PLOS ONE. Congratulations! Your manuscript is now being handed over to our production team.

Kind regards,

on behalf of

Dr. Zhanzhan Li

Academic Editor

PLOS ONE